# *Arabidopsis* BTB-A2s Play a Key Role in Drought Stress

**DOI:** 10.3390/biology13080561

**Published:** 2024-07-26

**Authors:** Guohua Cai, Yunxiao Zang, Zhongqian Wang, Shuoshuo Liu, Guodong Wang

**Affiliations:** School of Biological Sciences, Jining Medical University, Rizhao 276800, China; 19563349325@163.com (Y.Z.); w1258430940@163.com (Z.W.); 19563349606@163.com (S.L.)

**Keywords:** BTB-A2 protein, drought stress, ABA, stomata, *Arabidopsis*

## Abstract

**Simple Summary:**

Given that drought stress may particularly threaten plant survival and crop yields, plants have developed sophisticated adaptive strategies including drought tolerance, escape, and avoidance strategies, for dealing with drought stress. Abscisic acid has been widely acknowledged as a principal signaling molecule in plants responding to drought, reducing water loss by prompting stomatal closure while activating various stress-responsive genes. Broad-complex, Tramtrack, and Bric-à-brac (BTB) proteins in plants are important for plant growth and stress responses. The *Arabidopsis btb-a2.1/2/3* mutant confers drought tolerance by modulating plant growth parameters, physiology, and gene expression. Overall, AtBTB-A2s negatively regulate drought tolerance by suppressing stomatal closure and weakening ABA signaling. The results revealed the new physiological activity of AtBTB-A2s within *Arabidopsis* and the possible mechanism that mediates ABA-dependent signaling pathways during drought stress.

**Abstract:**

Drought stress significantly impacts plant growth, productivity, and yield, necessitating a swift fine-tuning of pathways for adaptation to harsh environmental conditions. This study explored the effects of *Arabidopsis* BTB-A2.1, BTB-A2.2, and BTB-A2.3, distinguished by their exclusive possession of the Broad-complex, Tramtrack, and Bric-à-brac (BTB) domain, on the negative regulation of drought stress mediated by abscisic acid (ABA) signaling. Promoter analysis revealed the presence of numerous ABA-responsive and drought stress-related *cis*-acting elements within the promoters of *AtBTB-A2.1*, *AtBTB-A2.2*, and *AtBTB-A2.3*. The *AtBTB-A2.1*, *AtBTB-A2.2*, and *AtBTB-A2.3* transcript abundances increased under drought and ABA induction according to qRT-PCR and GUS staining. Furthermore, the *Arabidopsis btb-a2.1/2/3* triple mutant exhibited enhanced drought tolerance, supporting the findings from the overexpression studies. Additionally, we detected a decrease in the stomatal aperture and water loss rate of the *Arabidopsis btb-a2.1/2/3* mutant, suggesting the involvement of these genes in repressing stomatal closure. Importantly, the ABA signaling-responsive gene levels within *Arabidopsis btb-a2.1/2/3* significantly increased compared with those in the wild type (WT) under drought stress. Based on such findings, *Arabidopsis* BTB-A2s negatively regulate drought stress via the ABA signaling pathway.

## 1. Introduction

Plants with a sessile life cycle must adapt to their surroundings to survive, facilitating individual growth and population reproduction. They often encounter various environmental challenges, such as extreme temperatures, salinity, drought, and pathogens, which can significantly affect crop productivity and quality. Among these challenges, drought poses a particularly critical threat, impacting the ability of plants to thrive on land [1]. Throughout the evolution of land plants, sophisticated adaptive strategies have emerged to cope with these environmental pressures. Specifically, in response to drought, plants have developed three main strategies: drought tolerance, escape, and avoidance [2,3,4]. Drought tolerance involves the maintenance of growth using a limited water content during drought via mechanisms such as scavenging reactive oxygen species (ROS), adjusting osmotic balance, and activating stress-related genes. Drought escape entails accelerating the plant’s life cycle to complete reproduction before stress becomes too severe. Drought avoidance focuses on minimizing water loss by quickly closing the stomata and delaying growth until favorable conditions return [2,4,5]. The most effective way among these adaptive measures is stomatal closure, which is achieved by curtailing water loss in a water-deficit environment [3,6,7].

Recent research advances have highlighted the significant impact of abscisic acid (ABA), a pivotal plant hormone and stress regulator, on regulating plant growth and adapting to stressors. Its effects include various crucial functions, including stomatal closure, seed germination, responses to drought and cold, and defense against pathogens [8]. ABA significantly affects orchestrating stomatal closure, particularly upon drought conditions [9]. More specifically, drought stress rapidly promotes stomatal closure in a manner predominantly dependent on ABA. Research has explored the stimulation of ABA production and accumulation within guard cells upon water deficit signals. Additionally, active synthesis occurs in other tissues, such as roots and leaf vasculature, facilitating transport to guard cells [9,10,11]. ABA accumulation can be recognized by specific receptors known as pyrabactin resistance (PYR)/PYR1-like (PYL)/regulatory components of ABA receptor (RCAR) receptors. These receptors deactivate clade A type 2C phosphatases (PP2C), subsequently activating downstream SNF1-related protein kinase 2.2 (SnRK2.2), SnRK2.3, and SnRK2.6 [12,13,14]. Among those members of the module, SnRK2s are important for positively accommodating ABA signal transduction to turn ABA signals on or off. Activated SnRK2s can then regulate stomatal closure to cope with water deficit by activating and phosphorylating channel proteins like SLAC1, KAT1, and numerous transcription factors such as ABI5, ABF2/AREB1, ABF3, and ABF4/AREB2 [15,16,17,18].

Broad-complex, Tramtrack, and Bric-à-brac/poxvirus and zinc finger (BTB/POZ) proteins, distinguished by approximately 120 conserved residues at their N-terminus, also referred to as the BTB domain, have been extensively investigated in eukaryotes [19,20]. Typically, these proteins act as potential substrate adaptors for CUL3 through the BTB domain and for substrate proteins via another protein–protein interaction domain [19]. Exploring the gene family of BTB domain-containing proteins across diverse plant varieties, including *Arabidopsis*, tomato, rice, corn, peach, and cucumber, can highlight their diverse biological functions [20,21,22,23,24,25]. In various biological processes, including chromatin organization, transcriptional regulation, cytoskeletal modulation, and protein degradation, BTB domain-containing proteins predominantly influence plant growth and environmental stress responses. Recent investigations have revealed their significance in plant stress responses. For instance, in *Arabidopsis*, non-expressers of pathogenesis-related genes NPR1, NPR3, and NPR4, which are salicylic acid receptors, have all been recognized as BTB/POZ proteins, exerting contrasting effects on the transcriptional control of the salicylic acid-mediated expression of defense genes [20,26]. Moreover, a tobacco BTB/POZ domain E3 ligase protein, POB1, restrains hypersensitive response programmed cell death (HR-PCD) in plant innate immune responses by facilitating the ubiquitin degradation of PUB17 [27]. In cucumber, salt stress markedly decreases the expression of the *CsBT1* gene [25]. Moreover, the sweet potato *IbBT4* gene confers drought tolerance [28]. OsMBTB32 enhances rice growth upon salt stress by interacting with OsCUL1s [29].

Previously, our research revealed that AtBTB-A2 proteins, expressed within the guard cell cytoplasm and nucleus, can interact with SnRK2.3 while influencing its stability, thereby impacting seed germination. Additionally, they also interact with SnRK2.6 [30]. According to the above results, it was hypothesized that AtBTB-A2 proteins were probably important for regulating responses to drought stress, a significant environmental challenge for plants. To explore this hypothesis further, we investigated the biological functions and mechanisms of AtBTB-A2 proteins associated with drought stress. Our results indicate that ABA and drought stress alter the transcript abundances of *AtBTB-A2.1*, *AtBTB-A2.2*, and *AtBTB-A2.3*. The genetic data suggest that AtBTB-A2 proteins function in drought stress responses, thereby influencing ABA signaling responsive gene expression in the meantime of modulating drought tolerance. Therefore, our findings suggest that the negative regulatory roles of AtBTB-A2s in ABA responses lead to reduced drought stress tolerance, providing insights into their function within the intricate network of drought stress adaptation.

## 2. Materials and Methods

### 2.1. Plant Materials and Growth Conditions

The plants used were generated in an *Arabidopsis* Columbia-0 (Col-0) background. The generation and identification of *btb-a2.1/2/3* triple mutant and *AtBTB-A2* overexpression lines have been detailed previously [30]. Transgenic plants were cultivated to the homozygous (T3) generation through selection with 10 µg/mL BASTA (Sigma-Aldrich, Milwaukee, WI, USA).

For the plant propagation and growth experiments, 75% ethanol was added to sterilize the surface of the *Arabidopsis* seeds for 5 min, followed by rinsing thrice using sterile water before sowing on half-strength Murashige and Skoog (MS) Phytoagar media that contained 1% (*w*/*v*) sucrose (pH 5.8). Following stratification for 3 days under 4 °C, we transferred the seeds to a growth incubator to achieve germination and growth with 16 h/8 h light/dark conditions at 22 °C. In soil culture, we transferred 10-day-old plants into nutrient-rich soil (Pindstrup Mosebrug, Denmark) for growth within the greenhouse at a light intensity of 150 µmol/m^2^/s, a temperature of 22 °C, and a light/dark cycle of 16 h/8 h.

### 2.2. Drought Assay

To ensure consistent soil moisture, each pot, filled with an equal amount of soil, was saturated with water. Sixteen 7-day-old plants displaying uniform growth, were planted in each small pot under identical humidity conditions. The control seedlings received watering every 3 days, while experimental seedlings underwent gradual drought stress through halting watering until the drought phenotype emerged, which was then followed by a 3-day period of watering.

### 2.3. GUS Staining

The construction and generation of *proBTB-A2.1*::GUS, *proBTB-A2.2*::GUS, and *proBTB-A2.3*::GUS transgenic lines have been described previously [30]. We conducted histochemical GUS staining according to previous methods [31]. The plant materials were soaked within GUS staining buffer (100 mM sodium phosphate, pH 7.0; 10 mM EDTA; 0.5 mM K_3_[Fe (CN)_6_]; 0.5 mM K_4_[Fe (CN)_6_]; 0.1% [vol/vol] Triton X-100) that contained 0.5 mM 5-bromo-4-chloro-3-indolyl-β-D-glucuronide (X-Gluc) and then vacuumed for 15 min. Following incubation for a 6 h period at 37 °C in the dark, we introduced 75% ethanol for decolorizing the samples several times and kept the samples in 95% ethanol.

### 2.4. Water Loss Determination

In the water loss assays, we detached the entire rosette from independent 4-week-old seedlings of each sample and placed them on Whatman filter paper on a laboratory bench (21–22 °C, about 40% relative humidity). Leaf fresh weight was measured and recorded at the indicated points, and fresh weight loss was analyzed. The test was replicated three times.

### 2.5. MDA Content Determination

We determined the MDA content in accordance with the method proposed by Kong [32]. One hundred milligrams of *Arabidopsis* leaves from normal or drought-stressed plants were ground in a cold mortar containing 0.1% (*w*/*v*) trichloroacetic acid (2 mL, TCA) as well as 8 mL of TCA for further grinding. After 10 min of centrifugation of the homogenate at 4000× *g* and 4 °C, 0.6% thiobarbituric acid (TBA) reagent (2 mL) was added to the supernatants (2 mL, with an equivalent amount of distilled water as a control). After a 15 min reaction in a boiling water bath, the reaction system was rapidly cooled, and centrifugated for a 10 min period at 5000× *g*. Supernatants were collected to measure the absorbance at wavelengths of 532 nm, 600 nm, and 450 nm. MDA contents were determined by the 155 mM^−1^ cm^−1^ extinction coefficient as described previously [33].

### 2.6. Relative Electric Conductivity (REC) Determination

The REC was measured according to the method proposed by Kong [32]. Briefly, the *Arabidopsis* leaves were rinsed with deionized water twice, and later, the surface moisture was removed using clean filter paper. Thirty leaf discs from each line were distributed into three clean tubes with ten discs in each tube. Deionized water (10 mL) was introduced into each tube containing leaf discs, which were then vacuumed for 20 min to remove air from the intercellular spaces and subsequently shocked for 1 h to measure the initial conductivity (S1) using a conductivity meter. Each tube was positioned for a 10 min period within the boiling water bath and later cooled to room temperature. The materials were equilibrated at ambient temperature for a 10 min period and shaken well for measuring final conductivity (S2). We recorded distilled water conductivity (S0) as a blank control. The REC could be determined as REC = (S1 − S0)/(S2 − S0) × 100.

### 2.7. Physiological Measurements of Guard Cells

The stomatal aperture test was carried out in line with Eisele’s method [34]. We first obtained the fourth expanded rosette leaves from one-month-old *Arabidopsis* plants. Following light incubation for a 2 h duration with stomatal opening buffer (10 mM MES, 10 mM KCl, 0.1 mM CaCl_2_, and pH 6.15) with the abaxial leaf surface downward to make the stomata as wide open as possible, the leaves were incubated within stomatal opening buffer that contained 10 µM ABA (Sigma-Aldrich, Milwaukee, WI, USA) for the indicated time points, and the epidermis on the back of the leaves was quickly peeled off. The stomata were observed under a microscope. ImageJ was utilized to determine the stromal width and length, while stomatal aperture indices were determined as the specific value of width compared with length.

### 2.8. RNA Isolation and qRT-PCR

RNA extraction was completed from *Arabidopsis* plants using TRIzol reagent (Invitrogen). First-strand cDNA was prepared with M-MLV reverse transcriptase (Promega, Madison, WI, USA) for RT-PCR with gene-specific primers. Thereafter, the Bio-Rad C1000 Thermal Cycler system was used for qRT-PCR with a SYBR Green I Master kit (Roche Diagnostics, Mannheim, Germany) for analyzing *AtBTB-A2s* expression within *Arabidopsis* seedlings under drought stress conditions and for examining ABA synthesis-related and ABA-responsive gene levels. *AtACTIN2* (AT3G18780) served as an internal control gene. The data were acquired from 3 biological replicates. Appendix A displays the primers used.

### 2.9. Statistical Analysis

All results in the current work were acquired from 3 separate experiments. The statistical analysis was conducted using Student’s *t*-test (* *p* < 0.05, ** *p* < 0.01).

## 3. Results

### 3.1. AtBTB-A2s Expression upon Drought Stress

Given that promoters can regulate gene expression, this work investigated the promoter region of *AtBTB-A2s*, which lies approximately 2000 bp upstream of the start codon. Several *cis*-acting elements related to stress responses were identified, including ABA-responsive elements, dehydration-responsive elements, MYB-binding sites critical for drought inducibility (MBS), and other elements (Appendix A). This prompted an investigation into whether *AtBTB-A2s* were induced by drought conditions. Firstly, according to qRT-PCR results, *AtBTB-A2.1*, *AtBTB-A2.2*, and *AtBTB-A2.3* transcripts increased in 7-day-old seedlings at specific time points under drought stress (Figure 1a). For characterizing the *AtBTB-A2’s* spatial tissue expression pattern and regulation, *proBTB-A2.1*::GUS, *proBTB-A2.2*::GUS, and *proBTB-A2.3*::GUS transgenic plants were constructed and generated. Subsequent GUS staining analysis further demonstrated that the *BTB-A2.1* expression level in *proBTB-A2.1*::GUS transgenic plants increased in both leaves and stems under drought conditions. Similarly, *BTB-A2.2* expression within leaves and roots in *proBTB-A2.2*::GUS transgenic plants increased under drought stress. *BTB-A2.3* expression within leaves, stems, and roots in *proBTB-A2.3*::GUS transgenic plants increased under drought stress (Figure 1b). These findings underscore that *AtBTB-A2s* are important for the drought stress response in *Arabidopsis*.

### 3.2. Mutation of AtBTB-A2s Enhances Drought Tolerance in Arabidopsis

Having established the induction of *AtBTB-A2.1*, *AtBTB-A2.2*, and *AtBTB-A2.3* by drought stress, we investigated the role of AtBTB-A2s in the drought response using the *Arabidopsis* T-DNA insertion triple mutant *btb-a2.1/2/3*, as described in our previous study [30]. Under normal conditions, the growth patterns of the wild type and the *btb-a2.1/2/3* mutant were similar. However, relative to the *btb-a2.1/2/3* mutant plants, the wild type exhibited significant wilting and decay due to severe drought stress, and the *btb-a2.1/2/3* mutant plants exhibited improved growth and some leaf curling (Figure 2a). Following a 3-day recovery period, the wild-type plants partially recovered, whereas the *btb-a2.1/2/3* mutant exhibited notable resilience (Figure 2a). Based on the number of surviving plants after recovery, the survival rates were approximately 27% for the wild type and 68.8% for the mutant *btb-a2.1/2/3* (Figure 2b).

Meanwhile, the *btb-a2.1/2/3*-treated leaves subjected to the isolated leaf assays in 15% PEG-6000 solution showed less severe purple discoloration than did the wild-type leaves after 12 h. Trypan blue staining is used as an indicator of cell death. The leaves from the wild-type plants had deeper staining compared with those from the *btb-a2.1/2/3* mutant plants, suggesting a greater cell death degree within wild-type leaves than mutant leaves (Figure 2c).

Under stress conditions such as drought, cold, or salt stress, the relative electrical conductivity (REC) increases due to a reduction in membrane integrity, and the malondialdehyde (MDA) content increases because of membrane system damage [35]. Therefore, we measured the MDA content and REC, which serve as important indicators for evaluating drought resistance in plants. The MDA content and the REC did not show notable differences in the wild type compared with mutant plants under normal watering conditions. Although the MDA content and REC were elevated within the water-deficit treatment group, these indicators were significantly lower in the *btb-a2.1/2/3* treatment group compared with the wild-type group (Figure 2d,e). Based on the above findings, the *Arabidopsis btb-a2.1/2/3* mutant enhances drought resistance.

### 3.3. AtBTB-A2s Expression after ABA Treatment

In plants, ABA is a pivotal hormone that orchestrates many physiological responses under stress conditions, notably extreme temperatures, salinity, and drought [36]. Previous investigations have indicated that AtBTB-A2s are negative modulators of ABA signal transduction in the seed germination phase in *Arabidopsis* [30]. Thus, we delved into assessing the expression levels of *AtBTB-A2s* in response to ABA treatment using GUS staining. Upon exposure to 50 µM ABA for 6 h, a marked increase in *AtBTB-A2.1* expression was observed in the leaves and stems of 15-day-old *proBTB-A2.1*::GUS transgenic seedlings. Similarly, the *AtBTB-A2.2* and *AtBTB-A2.3* expression markedly surged in the leaves, and in the leaves and roots, separately, of 15-day-old *proBTB-A2.2*::GUS and *proBTB-A2.3*::GUS transgenic seedlings (Figure 3). These findings suggest that ABA induces *AtBTB-A2.1*, *AtBTB-A2.2*, and *AtBTB-A2.3* expression.

### 3.4. Arabidopsis btb-a2.1/2/3 Reduced Water Loss under Drought Stress and Regulated ABA-Mediated Stomatal Closure

Evolution has equipped plants with strategies to combat drought stress, such as minimizing water loss to maintain limited water. Therefore, we assessed the water content upon drought stress. The wild-type plants had decreased relative water content relative to triple mutant *btb-a2.1/2/3*, indicating a more rapid water loss rate in the wild type compared with triple mutant *btb-a2.1/2/3* (Figure 4a). According to these results, the *btb-a2.1/2/3* mutant might confer drought tolerance by reducing the transpiration rate.

Plants reduce transpiration by closing their stomata, allowing them to survive under drought conditions. ABA facilitates stomatal closure or impedes stomatal opening, consequently minimizing water loss in aerial plant tissues [37]. We examined the dynamics of stomatal opening and closing after ABA treatment. According to our results, the stomata of the mutant plants closed at an increased rate after ABA treatment relative to those of the WT plants (Figure 4b,c). Consequently, we speculate that differences in leaf water loss rates may be the main cause of differences in drought tolerance, while *btb-a2.1/2/3* may exert their effects by promoting stomatal closure upon drought stress.

### 3.5. Sensitivity of AtBTB-A2 Overexpression Lines to Drought Stress

Given the enhanced drought resistance observed from the *btb-a2.1/2/3* mutant, we investigated the drought response of *AtBTB-A2.1*, *AtBTB-A2.2*, and *AtBTB-A2.3* overexpression lines. As shown in Figure 5a, under normal conditions, wild-type plants showed similar growth to overexpression plants. Upon drought conditions, the overexpression lines of *AtBTB-A2.1*, *AtBTB-A2.2*, and *AtBTB-A2.3* exhibited pronounced wilting and decay, different from the healthier growth observed from wild-type plants. Following a 3-day recovery period, most of the overexpression plants showed minimal recovery, while the wild-type plants recovered better. According to survival statistics, the survival rate of the wild type was around 64.6%, whereas the survival rates of the *AtBTB-A2.1*, *AtBTB-A2.2*, and *AtBTB-A2.3* overexpression lines were about 41.7%, 25%, and 20.8%, respectively (Figure 5b). Consistently, detached leaf assays further confirmed more rapid water loss in overexpression plants than in wild-type plants, resulting in earlier wilting (Figure 5c). Finally, an analysis of stomatal apertures revealed larger openings among the overexpression plants relative to the WT plants after ABA treatment (Figure 5d,e). Thus, the overexpression of *AtBTB-A2.1*, *AtBTB-A2.2*, and *AtBTB-A2.3* enhanced drought sensitivity through increasing stomatal aperture.

### 3.6. AtBTB-A2s Are Likely Related to Drought Stress Dependent on ABA Signaling Pathways

Previous research has demonstrated that AtBTB-A2s interact with SnRK2.6, which is important for the drought stress response [30]. As the *btb-a2.1/2/3* mutant exhibited drought resistance, this prompted us to investigate the involvement of AtBTB-A2s during ABA-mediated drought stress responses. Initially, it was proposed that the *NCED3* gene, which encodes critical and rate-limiting enzymes related to ABA biosynthesis, could regulate water stress through ABA accumulation [38]. The *AtBTB-A2.1*, *AtBTB-A2.2*, and *AtBTB-A2.3* expression was analyzed in the wild type and the *atnced3* mutant, revealing no significant difference between them (Appendix A). Subsequently, we examined the ABA biosynthesis-related gene (*AtAAO3*, *AtABA1*, *AtABA3*, and *AtNCED3*) expression levels in the wild type and the *btb-a2.1/2/3* mutant before and after exposure to drought conditions. Although drought stress increased the gene levels within both WT and mutant plants, no notable differences were detected (Appendix A), implying that the expression of *AtBTB-A2s* may be not related to changes in endogenous ABA expression during drought stress. Finally, we detected the levels of ABA signaling-responsive genes (*AtABI5*, *AtRAB18*, *AtRD29A*, and *AtRD9B*) through qRT-PCR. We discovered that after drought stress, the levels of ABA signaling-responsive genes markedly increased, but the up-regulated expression was more significant in the *btb-a2.1/2/3* mutant compared with the WT (Figure 6). According to the above findings, AtBTB-A2s might exert a negative regulation on the drought stress response in a manner dependent on the ABA signaling pathway.

## 4. Discussion

Upon environmental stresses, dramatic changes in physiology, metabolism, and especially gene expression often occur [39,40]. In plants, BTB proteins exert vital effects on plant growth as well as stress responses. AtSIBP1, the potential substrate receptor for the CRL3 complex, positively modulates the salt stress response by suppressing ROS accumulation [41]. MdBT2 can negatively regulate drought stress response by enhancing the transcription factor MdNAC143 degradation in apples [42]. AtBPH1, a novel substrate receptor of CRL3, can exert a negative regulation of ABA-induced cellular responses, like drought stress and seed germination [43]. Nonetheless, the exact effects of numerous BTB genes in plants, particularly stress responses, remain unknown. This study discovered that three *Arabidopsis* BTB-A2 subfamily genes negatively regulate drought resistance by suppressing stomatal closure and weakening ABA signaling. This research revealed the new physiological activity of AtBTB-A2s within *Arabidopsis* and revealed the possible mechanism that mediates ABA-dependent signaling pathways during drought stress.

Here, *AtBTB-A2s* transcript levels rapidly increased upon drought stress, suggesting that *AtBTB-A2s* may have the potential to function under drought stress. Considering that we previously reported that AtBTB-A2.1, AtBTB-A2.2, and AtBTB-A2.3 are similar in terms of their localization and expression patterns and that the resulting polymers redundantly participate in biological activities [30], we constructed triple mutant and *AtBTB-A2s* overexpressing transgenic plants harboring these three proteins and screened their response to drought stress. According to the results, compared with the WT plants, the *btb-a2.1/2/3* mutant plants at the adult stage exhibited increased tolerance to drought stress (Figure 2a,b). Furthermore, this conclusion was supported by the results showing that *btb-a2.1/2/3* grew better than the wild type under PEG treatment, and the REL, MDA content, and degree of cell death in *btb-a2.1/2/3* decreased relative to those in the WT (Figure 2c–e). Conversely, the overexpression transgenic plants exhibited increased sensitivity to drought stress relative to the wild-type plants (Figure 5). Based on the above results, AtBTB-A2s may negatively regulate tolerance to drought stress.

Plants react to water-deficit stress via intricate signaling pathways, resulting in stomatal closure [44]. ABA is widely acknowledged as the principal signaling molecule in the drought response of plants, reducing water loss by prompting stomatal closure and activating various stress-responsive genes [45,46]. Initially, we observed a robust induction of *AtBTB-A2s* by external ABA according to GUS staining (Figure 3). Additionally, AtBTB-A2.1, AtBTB-A2.2, and AtBTB-A2.3 interact with a component in the ABA signaling pathway, SnRK2.6 [30]. Third, AtBTB-A2s inhibited stomatal closure via the ABA-dependent mode. Fourth, the *btb-a2.1/2/3* mutant impacted ABA-responsive gene expression, but not ABA synthesis-related gene levels under drought stress (Figure 6). These results confirm the ability of the *btb-a2.1/2/3* mutant to reduce plant water loss rate upon drought stress (Figure 4). According to the above findings, AtBTB-A2s confer sensitivity to drought stress by reducing water retention via the ABA-dependent signaling pathway.

Multiple plant BTB protein family members, containing the conserved BTB motif, can regulate substrate recognition while acting as substrate receptors for the Cul3 E3 ubiquitin ligase complex [47]. Our data demonstrated that *Arabidopsis* BTB-A2s negatively affected the response of plants to drought stress via the ABA signaling-dependent manner. Moreover, several ABA signaling transducers are regulated via ubiquitination through their impact on protein localization, activity, interaction ability, and assembly, resulting in changes in ABA-mediated cellular events [48]. We succeeded in identifying the potential target protein of AtBTB-A2s, SnRK2.6, which is also called open stomata 1 (OST1) and is a serine/threonine protein kinase with important effects on connecting ABA receptor complexes with downstream components, including responsive factors (RD29A, RD29B, RAB18, ABI5, and so on) and anion channels, thus regulating stomatal aperture in the stress response. Previous studies have indicated that the up-regulated transcript abundances of stress-responsive genes indicate that plants can manage drought stress [49]. Additionally, under drought stress, the up-regulation of certain responsive factors, like the ABA signaling-related genes *ABI5*, *RD29A*, and *RD29B*, positively confer drought tolerance to plants [50,51,52,53]. Some stress-responsive players have been discovered, but the mechanisms that regulate pathway activation duration and amplitude, which have a critical effect on stress adaptation, are largely unclear. Overall, our findings suggest that AtBTB-A2s likely participate in negatively regulating ABA signaling. ABA up-regulated AtBTB-A2 expression, thereby regulating SnRK2.6 during drought stress (Figure 7). However, how AtBTB-A2s affect SnRK2.6 upon drought stress needs further investigation to clarify the underlying mechanisms involved.

## 5. Conclusions

This study used reverse genetics methods to reveal a novel biological role and potential mechanism of *Arabidopsis BTB-A2* genes in drought tolerance. According to the results, AtBTB-A2s exert a negative effect on the drought stress response. The present work sheds novel light on the physiological effects of BTB-A2s in *Arabidopsis* and offers further insight into the molecular mechanisms involved in ABA-dependent signaling pathways during drought stress.

## Figures and Tables

**Figure 1 biology-13-00561-f001:**
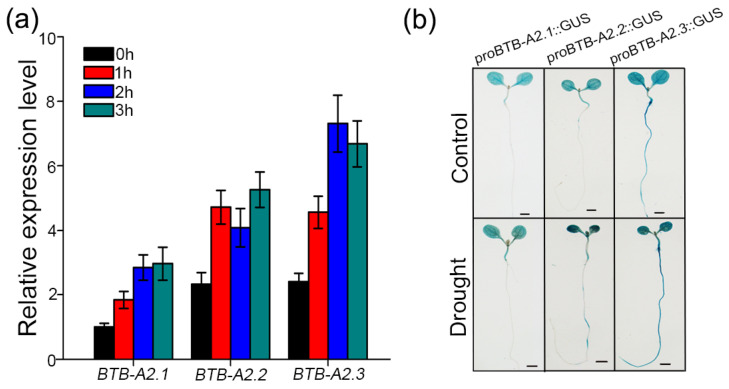
Effect of drought on *AtBTB-A2.1*, *AtBTB-A2.2*, and *AtBTB-A2.3* expression patterns. (**a**) Effect of drought stress on *AtBTB-A2.1*, *AtBTB-A2.2*, and *AtBTB-A2.3* expression was determined through qRT-PCR. Seven-day-old wild-type plants experienced drought stress for 0, 1, 2, or 3 h. *AtACTIN2* gene served as internal control. (**b**) Changes in 7-day-old *proBTB-A2.1*::GUS, *proBTB-A2.2*::GUS, and *proBTB-A2.3*::GUS transgenic seedlings prior to and following 2 h drought stress were determined by GUS staining. Upper lane represents transgenic plants without treatment, and second lane represents transgenic plants after 2 h of drought treatment. Bar = 1 mm.

**Figure 2 biology-13-00561-f002:**
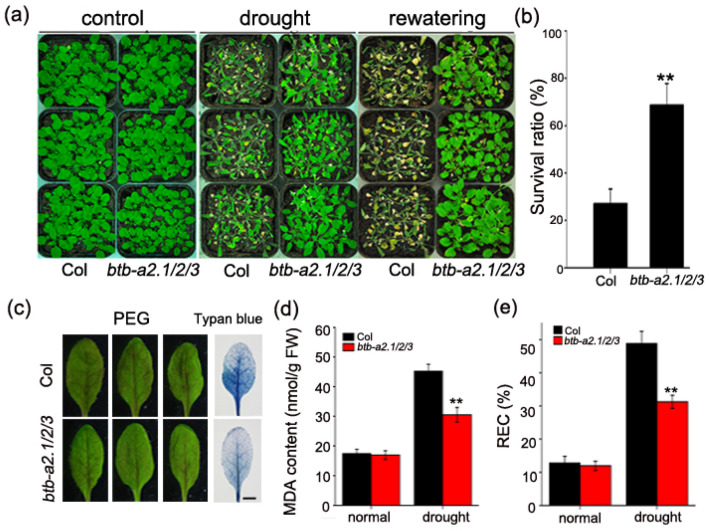
*Arabidopsis btb-a2.1/2/3* mutant exhibited enhanced drought tolerance. (**a**) Phenotypes of 4-week-old wild-type plants and *btb-a2.1/2/3* mutants with/without drought stress for 4-week days and after 3-day rehydration. (**b**) Survival ratio assay. (**c**) Phenotypes of detached leaves from 3-week-old wild-type plants and *btb-a2.1/2/3* mutant plants soaked in 15% PEG-6000. Trypan blue-stained leaves from wild type and *btb-a2.1/2/3* plants after PEG-6000 treatment. Bar = 2 mm. (**d**–**e**) Measurement of MDA content and REC of wild type and *btb-a2.1/2/3* mutant plants with/without drought conditions. Results are shown as mean ± SD from 3 biological replicates. ** *p* < 0.01 stood for statistical significance in wild type compared with *btb-a2.1/2/3* plants.

**Figure 3 biology-13-00561-f003:**
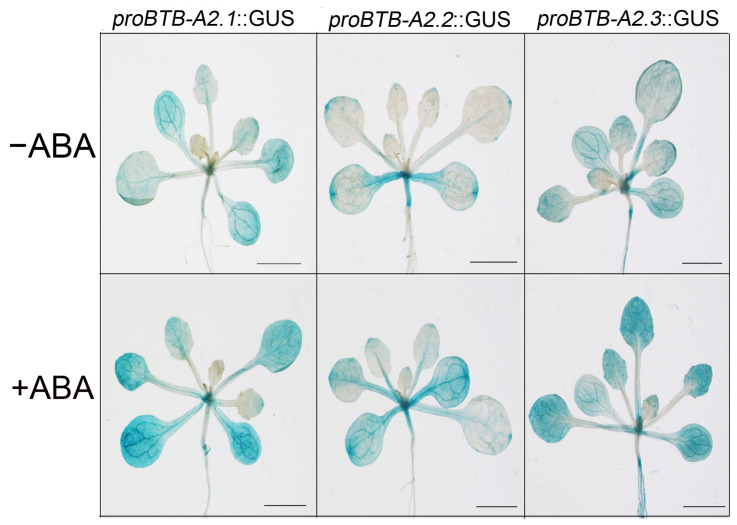
Effect of ABA on *AtBTB-A2.1*, *AtBTB-A2.2*, and *AtBTB-A2.3* expression patterns. Changes in *proBTB-A2.1*::GUS, *proBTB-A2.2*::GUS, and *proBTB-A2.3*::GUS among 15-day-old transgenic plants before and after 6 h treatment with 50 µM ABA were determined by GUS staining. Bar = 2 mm.

**Figure 4 biology-13-00561-f004:**
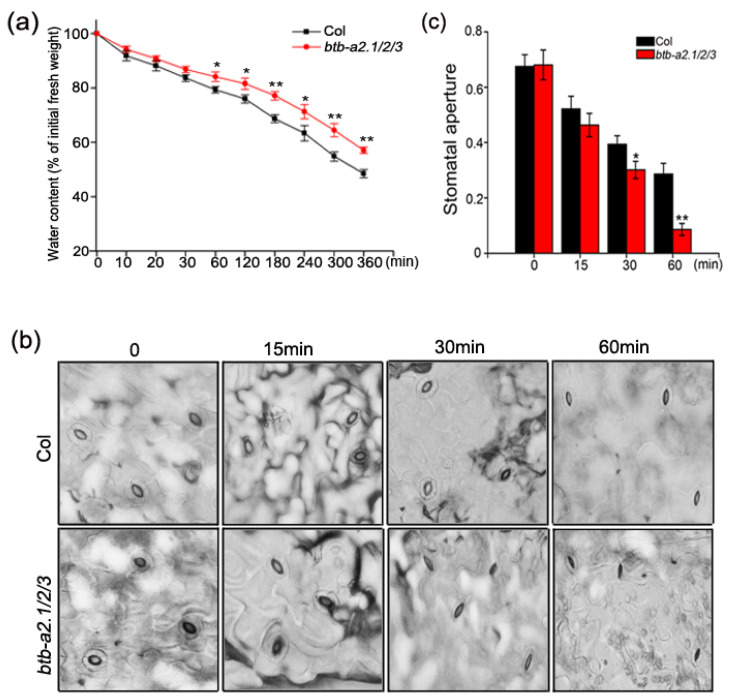
Water loss of the *btb-a2.1/2/3* mutant decreased upon drought stress through ABA-induced stomatal closure. (**a**) Relative water content measured in detached leaves of control and mutant plants. (**b**) Stomatal opening of the WT and triple mutant plants at different time periods with and without 10 µM ABA. (**c**) Stomatal aperture measurements of the WT and triple mutant plants at varying time periods after a 10 µM ABA treatment. Approximately 100 stomas were analyzed for each line at each time point. The results are shown as mean ± SD from 3 biological replicates. * *p* < 0.05 and ** *p* < 0.01 stood for statistical significance in the wild type compared with *btb-a2.1/2/3* plants.

**Figure 5 biology-13-00561-f005:**
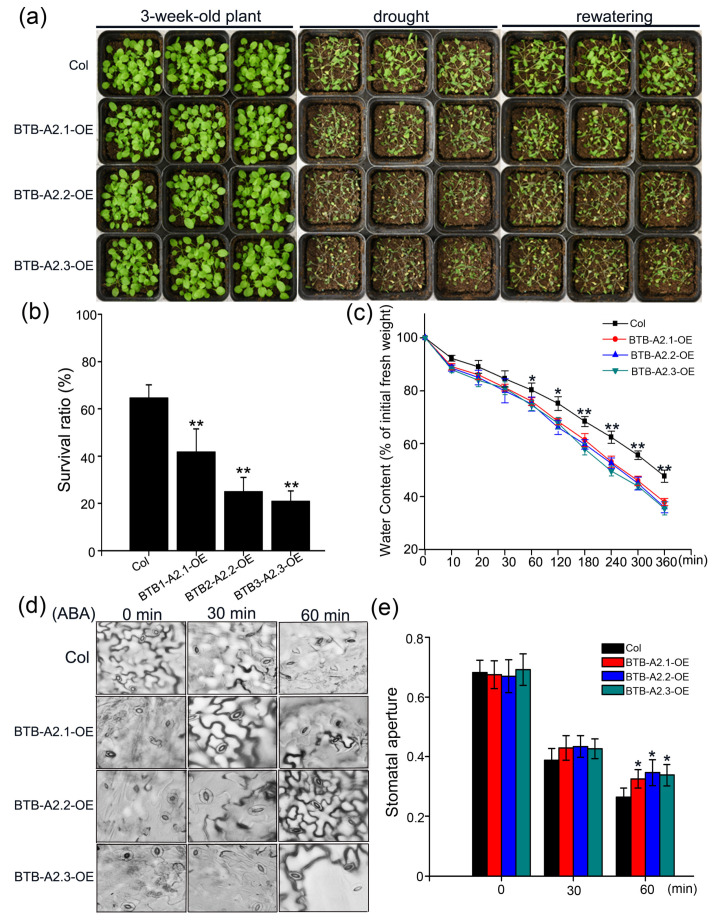
*AtBTB-A2.1*, *AtBTB-A2.2*, and *AtBTB-A2.3* overexpressing plants exhibited enhanced drought sensitivity. (**a**) WT and overexpression transgenic plant phenotypes upon normal and drought stress conditions for 3-week days and after 3-day rewatering. (**b**) Survival ratio assay. (**c**) Measurement of relative water content by detached leaf samples in control and transgenic plants under drought conditions. (**d**) Stomatal opening of control and transgenic plants at specific time periods with/without 10 µM ABA. (**e**) Stomatal aperture measurements of control and transgenic plants at specific time periods under 10 µM ABA treatment. Approximately 100 stomas were analyzed for each line at each time point. The results are indicated by mean ± SD from 3 biological replicates. * *p* < 0.05 and ** *p* < 0.01 stood for statistical significance in the wild type compared with *btb-a2.1/2/3* plants.

**Figure 6 biology-13-00561-f006:**
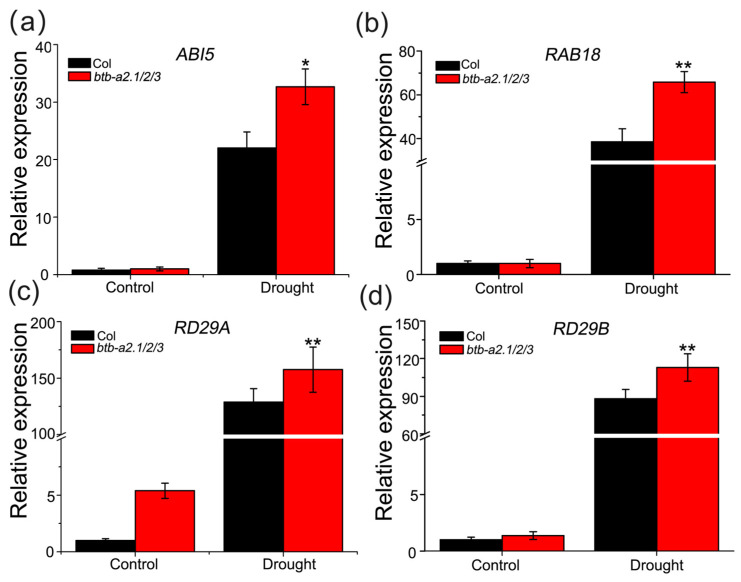
qRT-PCR was carried out to measure ABA signal-responsive gene levels upon drought stress. (**a**–**d**) The expression levels of *AtABI5*, *AtRAB18*, *AtRD29A*, and *AtRD29D* in the WT and *btb-a2.1/2/3* mutant upon normal and drought conditions. The *AtACTIN2* gene served as the endogenous reference. The results are presented as mean ± SD from 3 biological replicates. * *p* < 0.05 and ** *p* < 0.01 stood for statistical significance in the wild type compared with *btb-a2.1/2/3* plants.

**Figure 7 biology-13-00561-f007:**
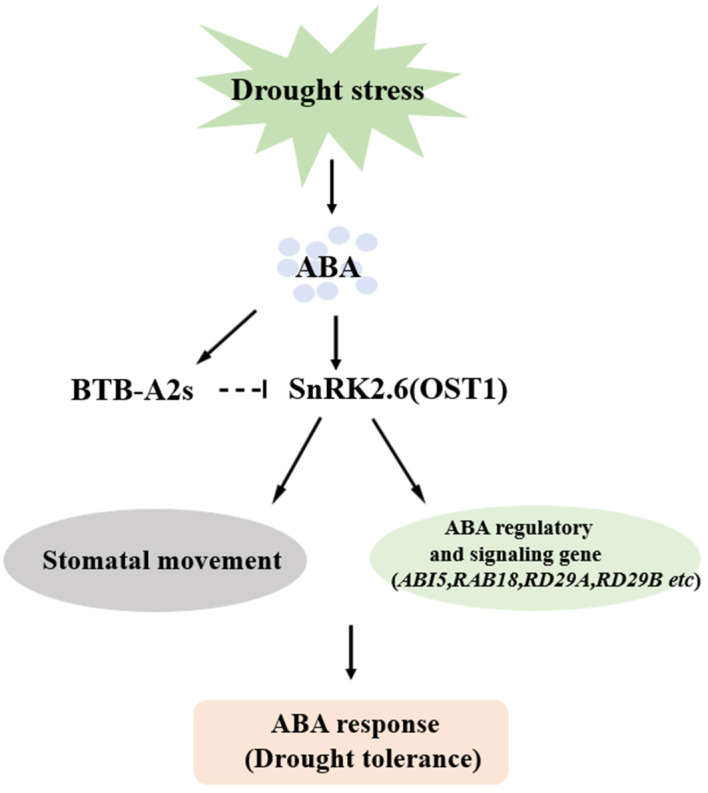
Proposed model for negative effects of AtBTB-A2s under drought stress.

## Data Availability

Data associated with the present article are available from this article and Appendix A.

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
