# Peer review of "Arabidopsis BTB-A2s Play a Key Role in Drought Stress"

_biology, 2024, doi:10.3390/biology13080561_

Round 1

Reviewer 1 Report

Comments and Suggestions for Authors

I am pleased to review the manuscript titled "AtBTB-A2s Negatively Regulate Drought Stress through the 2 Abscisic Acid Signaling Pathway in Arabidopsis".

The authors examined the function of Arabidopsis BTB-A2s in drought stress and showed how ABA influenced the BTB-2As function.

I have the following observations.

1. The English language in the paper should be revised, many errors and the use of AI or software should be avoided.

2. The possession mark in line 11 and line 41 "Plants'" is not properly used.

3. in the introduction line 37  "In the natural world" is a redundancy.

4. In lines 107 to 109, the sentence structure is incorrect and unclear.

5. Line 149 "The supernatants were gathered" should be "The supernatants were collected"

6. lines 187 to 203 should be completely revised to express the intended scientific message

7. lines 231 to 232 also need to be revised "Under unfavorable conditions" should be "Under stress conditions" 

8. the mutant used in the study is a triple mutant, to get a clearer picture of the phenotype of single mutants should be shown to know which among the btb-a2.1/2/3 is stronger.

9. The result "3.4. Arabidopsis btb-a2.1/2/3 Reduced Water Loss under Drought Stress and Regulated ABA- 268 Mediated Stomatal Closure" is not clear and unorganized. Please rewrite the whole result.

10. Lines 301 to 302 " in stark contrast to the significant resilience observed in the wild type" What is stark contrast? the sentence should be concise and precise.

11. the model in Figure 7 can also be revised for a better understanding of the AtBTB-A2s.

Comments on the Quality of English Language

The English language in the paper should be revised and AI should be avoided.

Reviewer 2 Report

Comments and Suggestions for Authors

The authors study the role of ABA-regulated genes (BTB-A2.1, BTB-A2.2 and BTB-A2.3) in drought tolerance of A. thaliana plants. Using triple mutants for the studied genes, it was convincingly shown that these genes down-regulate plant drought tolerance. This could be partly due to the fact that these proteins prevented the closure of the stomata, resulting in water loss. Transgenic plants containing promoter regions of the studied genes were obtained and analyzed.  The assumed role of the proteins was confirmed by the results of the analysis of transgenic plants overexpressing the studied genes. The BTB-A2.1, BTB-A2.2, and BTB-A2.3 genes are regulated by water deficiency and ABA. The authors believe that BTB-A2 proteins act through the ABA signaling pathway.

 Many good results are obtained, but there are some questions and comments.

1- The authors state in the title of the paper that "AtBTB-A2s Negatively Regulate Drought Stress through the Abscisic Acid Signaling Pathway", but they do not have enough evidence for this conclusion. 

(a) "We observed robust induction of AtBTB-A2s by external ABA according to GUS staining". This suggests not ABA signaling but possibly transcriptional regulation.

(b) "AtBTB-A2s inhibited stomatal closure in an ABA-dependent manner" A very indirect indication of ABA signaling.

(c) "btb-a2.1/2/3 mutant affected the levels of ABA-responsive genes". So what of this?

d) The most serious confirmation of the authors' ideas is the earlier results "AtBTB-383 A2.1, AtBTB-A2.2, and AtBTB-A2.3 interact with a component of the ABA signaling path-384 pathway, SnRK2.6"

In my opinion, the authors should change the title of the article, giving a touch of suggestiveness.

 2.      2.2. Drought Assay.    It is not clear how and for how long the drought was created.  124-125 "...drought stress by halting watering until the drought phenotype emerged..." Why is the exact timing of drought creation not stated. With the authors' approach, each experiment could have a different drought duration?

3.      When studying drought, it is necessary to characterize the water status of plants.

4.      2.3 GUS Staining. Constructs with promoters of the studied genes (in proBTB-A2.1::GUS, proBTB-A2.2::GUS, and proBTB-A2.3::GUS ) and constructs overexpressing the genes of the studied proteins and the resulting transgenic plants should be at least briefly described.

5.      170 and 256 How the optimal ABA concentration of 10 and 50 µM was determined.  50 µM is a very high concentration.

6.      189-190 "...located around 2000 bp upstream of 189 the start codon..." meaning transcription initiation site or translation site?

7.      Fig 2c does not show any purple coloration of the leaves. 246 "...(d-e) Measurement of the MDA content ..." No part of the figure is labeled (d). It is difficult to compare the results presented in Figures 1a and 1b.

 8.      I may be wrong, but Figure 4b and similar ones showing the stomata are of insufficient quality.

9.      374 "...btb-a2.1/2/3 grew better than the wild type under PEG treatment...". If I understand correctly, the wild type and mutant plants did not grow with PEG6000, but the leaves cut from the plants were soaked in PEG solution, which follows from the figure caption. Lines 244-245 "Phenotypes of detached leaves of 3-week-old wild-type plants and 244 btb-a2.1/2/3 mutant plants soaked in 15% PEG-6000" 

10.    195-196 "...that the mRNA expression of AtBTB-A2.1, At-195 BTB-A2.2, and AtBTB-A2.3 increased in 7-day-old seedlings..." Expression of genes, not mRNAs. In addition, it is better to speak of transcripts rather than mRNAs, since amplification can also occur from mRNA precursors.

11.    276 "...ABA, functioning as a plant hormone..." it was known more than half a century ago that ABA is a plant hormone and it makes no sense to emphasize this again.

12.    337-338 AtABI5, AtRAB18, AtRD29A, AtRD9B it would be useful to explain why these particular genes are taken for analysis.

Round 2

Reviewer 1 Report

Comments and Suggestions for Authors

I have no further comments, all my concerns are addressed.

Reviewer 2 Report

Comments and Suggestions for Authors

The authors have done some serious work on the text of the article. They have made significant revisions and have taken into account all my recommendations.  The article has improved and in this form I recommend that it be accepted for publication in the journal

All the best